# Numerical Study of Reinforced Aluminum Composites for Steering Knuckles in Last-Mile Electric Vehicles

Carlos Santana [1], Luis Reyes-Osorio [1,*], Jesus Orona-Hinojos [2], Lizbeth Huerta [2], Alfredo Rios [2] and Patricia Zambrano-Robledo [1]

1   Facultad de Ingeniería Mecánica y Eléctrica, Universidad Autónoma de Nuevo León, San Nicolás de los Garza 66455, Mexico; csantanad@uanl.edu.mx (C.S.); patricia.zambranor@uanl.edu.mx (P.Z.-R.)
2   DataSc Consultancy Group, Saltillo-Coahuila 25084, Mexico; jmorona70@gmail.com (J.O.-H.); lahuertal1990@gmail.com (L.H.); noguesmx@gmail.com (A.R.)
*   Correspondence: luis.reyessr@uanl.edu.mx; Tel.: +52-811-0165-901

**Abstract:** The steering knuckle is a critical component of the suspension and steering drive systems of electric vehicles. The electrification of last-mile vehicles presents a challenge in terms of cost, driving range and compensation of battery weight. This work presents a numerical methodology to evaluate 60XX series aluminum metal matrix composites (AMMCs) with reinforcement ceramic particles for steering knuckle components in medium heavy-duty last-mile cargo vehicles. The use of AMMCs provides lightweight knuckles with sufficient strength, stiffness and safety conditions for electrical vehicle cargo configurations. The numerical study includes three aluminum alloys, two AMMC alloys and an Al 6061-T6 alloy as reference materials. The medium-duty heavy vehicle class < 12 t, such as electrical vehicle cargo configurations, is considered for the numerical study (class 1–4). The maximum von Mises stress for class 4 AMMC alloys exceeds 350 MPa, limited by fracture toughness. The weight reduction is about 65% when compared with commercial cast iron. Moreover, Al 6061-T6 alloys exhibit stress values surpassing 300 MPa, constraining their suitability for heavier vehicles. The study proposes assessing the feasibility of implementing AMMC alloys in critical components like steering knuckles and suggests solutions to enhance conventional vehicle suspension systems and overcome associated challenges. It aims to serve as a lightweight design guide, offering insights into stress variations with differing load conditions across various cargo vehicles.

**Keywords:** steering knuckle; metal matrix composites; lightweight design; finite element method

## 1. Introduction

The electrification of vehicles for goods transportation is vital to reduce pollution and improve air quality in urban centers. Several manufacturers have recently introduced battery electric trucks, technically becoming electric heavy-duty trucks that are commercially viable. A study in Canada concluded that more than 65% of medium-duty trucks must be electrified by 2040 due the rigorous regulations for fuel-efficient diesel trucks [1]. Electrification potential studies are carried out considering the following specifications: (1) for medium and heavy rigid trucks, a battery capacity of 150–250−350 kWh must be implemented, while semitrailer and articulated trucks must hold 400–600−800 kW h of battery capacity; (2) based on the gravimetric density of batteries with 120–240−360 W h/kg, the weight is projected to be 850 kg for rigid trucks and 1700 kg for heavier trucks; and (3) a charging power of 22–50–150 kW with time of 8 h (overnight) and on-road recharging power of 50–150−250–400 kW with a time of 2 h during the day must be provided.

Heavy-duty vehicles for last-mile cargo distribution are potential alternatives to electrification due to the short journey lengths (less than 50 km) and fixed routes [2]. The lightweight design of suspension components plays a key role in vehicle efficiency. The suspension system is responsible for maintaining contact between the wheels and the road

surface, which is essential for stabilizing the vehicle body [3]. For commercial vehicles, the primary aim of the suspension system is to maximize the comfort of the driver and passengers by minimizing the amplitude of vertical accelerations by the occupants. In contrast, for racing purposes, the function of the suspension system is to enhance the performance of the vehicle and expand its dynamic range [4].

In MacPherson suspension systems, the steering knuckle connects the strut assembly to the lower ball joint. The steering knuckle serves as the pivot point for the steering system, enabling the wheels to turn. The spindle in steering knuckles plays a vital role in locating and supporting the inner and outer wheel bearings. Therefore, as an application, it can be stated that the knuckle joint in automotive design is responsible for holding the weight of the front axle as well as supporting and driving the rotation of the front wheel around the main pin, which plays a crucial role in steering [5]. Knuckle joint failure can result in serious consequences, such as the loss of steering control and vehicle stability, which can lead to accidents [6].

The steering knuckle requires a stiff and high-strength material to endure static and dynamic loads in the vehicle. Typically, steering knuckles are made of cast iron. However, new trends in lightweight vehicles incorporate aluminum alloys as a replacement for traditional cast iron. The manufacturing process of steering knuckles must consider various factors such as material choice, manufacturing method and machining operation. Metal matrix composites (MMCs) offer outstanding properties, helping to reduce knuckle weight without compromising performance. In order to maximize the qualities of the composite, a careful selection of a particular set of variables must be made since the mechanical properties of the composite material are highly dependent on the microstructural parameters of the matrix reinforcement system. In particular, the composition of the matrix or its thermal heat treatment, as well as the form, size, volume percentage and orientation of the reinforcing particles, must be carefully determined [7]. Aluminum metal matrix composite AMMCs are lightweight, highly corrosion-resistant and extremely durable. These characteristics allow AMMCs to be used in a variety of ways in the automotive, marine and aerospace sectors [8].

Diverse studies have been developed to evaluate the mechanical responses of steering knuckles under static and dynamic loadings. One of the challenges during the structural analysis of knuckle components is the difficulty of transferring data from specimens to the actual component. Due to the difference between the geometry and specifications of the component, which often differ from those of the analyzed specimen, it has been challenging to define a stress state or a notch factor. However, component testing considers the effects of material, fabrication parameters and geometrical characteristics, which work synergistically to produce the final component [9].

Numerical methods, such as the finite element method, allow the prediction of axial and shear stress that occur in the knuckle component under different load conditions. The experimental and numerical evaluation of dynamic loading conditions results in better efficiency of knuckle components [10]. Recently, some studies have been developed to evaluate the performance of steering knuckles using lightweight designs. Kim [11] performed a finite element static and modal analysis of a vehicle steering knuckle to optimize its shape and material in order to reduce its weight. Al 6061-T6 was found to be a suitable material in terms of dynamic conditions. The component optimization reduced the weight by 50%. The stress distribution of the knuckle was evaluated under four loading conditions. The change from an iron alloy to a lightweight Al alloy was verified in terms of maximum von Mises stress.

Chen et al. [12] developed a design and analysis of a steering knuckle for an electric vehicle. A finite element model was created based on the vehicle weight and suspension requirements. Three road classes were evaluated, reporting the maximum strength of the knuckle. Madhusudhanan et al. [13] performed an experimental and numerical study to validate the static strength of a steering knuckle made of cast iron. It was observed that a fracture occurred in the steering arm section. Vijayarangan et al. [14] determined

the knuckle strength under different driving conditions using a multibody approach. Aluminum reinforced with titanium carbide was selected to replace spheroidal graphite iron. The experimental and numerical results indicate that metal matrix composites are alternatives for steering knuckles, with a weight saving of about 55%. Kim et al. [15] performed a lightweight design of a steering knuckle replacing a cast iron alloy with an Al alloy. A structural optimization was performed, which reached a weight reduction of 60%. Tagade et al. [16] used a finite element study to optimize a steering knuckle made of Al alloy. A vehicle of 1480 kg was considered in the numerical study. The stress results were below the yield stress of the material, showing potential as an alternative material for steering knuckles. Sharma et al. [17] developed a static analysis and optimization of a knuckle using a finite element model. An Al 2011-T3 alloy was selected due to its low weight and tensile strength. The shape optimization achieved a significant reduction in weight for the knuckle. Srivastava et al. [18] performed an optimization procedure using a finite element model, which achieved a mass reduction of 19% without compromising the knuckle performance. Saravanan et al. [19] performed a numerical study to evaluate the multiaxial force acting on a knuckle considering three load cases. It highlights the importance of weight reduction in the automotive industry and topology optimization. In addition, Yadav et al. [20] described the design and numerical analysis of a knuckle by finite element analysis. An optimization study was performed to decrease the weight of the suspension component and improve the vehicle performance.

Recently, additive manufacturing has been explored for producing high-quality and reliable knuckles [21]. Kim et al. [22] performed a finite element study of an additive manufacture steering knuckle for an electric vehicle. An increment of stiffness of about 2.5 times compared with a previous design was observed. Gupta et al. [23] developed a structural optimization process of the front knuckles of a formula student prototype vehicle through finite element analysis. An Al 7075 alloy reinforced with silicon carbides was selected for the numerical study. It was observed that metal matrix composite alloys reduced the vehicle weight without compromising performance. In addition, Reza [24] developed a static analysis of an AMMC knuckle through a finite element model. A multi-dynamic model was performed considering a maximum loading value for the stress study. It was observed that an increment of volume of titanium carbide particles leads to an increment of steering knuckle strength.

The research highlights several significant research gaps in the field of AMMC alloys for steering knuckles, particularly in the context of electrified cargo vehicles, as follows:

1. Lack of Research on AMMC Alloys for Steering Knuckles: Despite existing research on AMMC alloys, there is a notable lack of studies specifically focused on their application in steering knuckles. This gap indicates a need for investigation of the suitability and performance of AMMC alloys specifically tailored for steering knuckles.

2. Absence of Studies on Electrified Cargo Vehicles: The research gap extends to the absence of studies evaluating AMMC alloys for steering knuckles of electrified cargo vehicles. Given the unique requirements and operating conditions of electrified vehicles, there is a need for research to assess the viability and effectiveness of AMMC alloys in this context.

3. Limited Understanding of Mechanical Properties and Forces: The significance of understanding the mechanical properties and forces that steering knuckles can withstand is underscored. This gap suggests a need for comprehensive investigations into the strength, durability and performance characteristics of steering knuckles, particularly when reinforced with lightweight AMMC alloys.

4. Lack of Numerical Studies on Dynamic Loading Conditions: Another research gap pertains to the scarcity of numerical studies considering various dynamic loading conditions, such as braking force, lateral force and steering force, applied to steering knuckles. Understanding the response of steering knuckles under different loading scenarios is essential for optimizing their design and performance.

5. Focus on Specific Vehicle Classes: The research focuses on the medium-duty heavy vehicle classes (<12 t), particularly electrified cargo vehicle configurations (class 1–4). This indicates a specific gap in research pertaining to the application of AMMC alloys in steering knuckles for these vehicle classes, highlighting the need for targeted investigations tailored to their requirements.

Addressing these research gaps through numerical studies and static stress analyses can facilitate the development of lightweight designs, propose solutions to enhance conventional vehicle suspension systems and overcome associated challenges in the context of electrified cargo vehicles.

The novelty of this work lies in its comprehensive approach to evaluating the strength of front steering knuckles reinforced by aluminum metal matrix composites through numerical analysis. Unlike previous research, this study considers various dynamic loading conditions, including braking force, lateral force and steering force, applied at different locations on the knuckle. Particularly groundbreaking is the focus on medium-duty heavy vehicle class configurations (<12 t), specifically electrified vehicle cargo setups (class 1–4), which have received limited attention in existing literature. Additionally, the incorporation of static stress analysis enables the investigation of lightweight designs and the proposal of solutions to enhance conventional vehicle suspension systems, addressing associated challenges. This holistic examination of AMMC-reinforced steering knuckles in the context of electrified cargo vehicles represents a novel contribution to the field, offering insights into optimizing performance and durability while advancing lightweight design principles.

## 2. Materials and Methods

A numerical study of front suspension steering knuckles was developed using the software Ansys 2024 R1. The numerical procedure includes the geometrical design and preparation of CAD geometries, the development of a static structural model under different loading scenarios (classes 1–4) and the comparison of two AMMC alloys. In addition, Al 6061- T6 alloy was explored as a reference material. The CAD model was imported to Ansys as an IGES format for discretization and finite element solution. Figure 1 presents the principal parts of the front steering knuckle with a length and width of 0.51 m and 0.23 m, respectively. In Figure 1a, the upper and lower arms, brake mounting, tie rod and stub hole are observed. The stub hole houses the bearing of the wheel and transfers different loads during acceleration, braking and bumping.

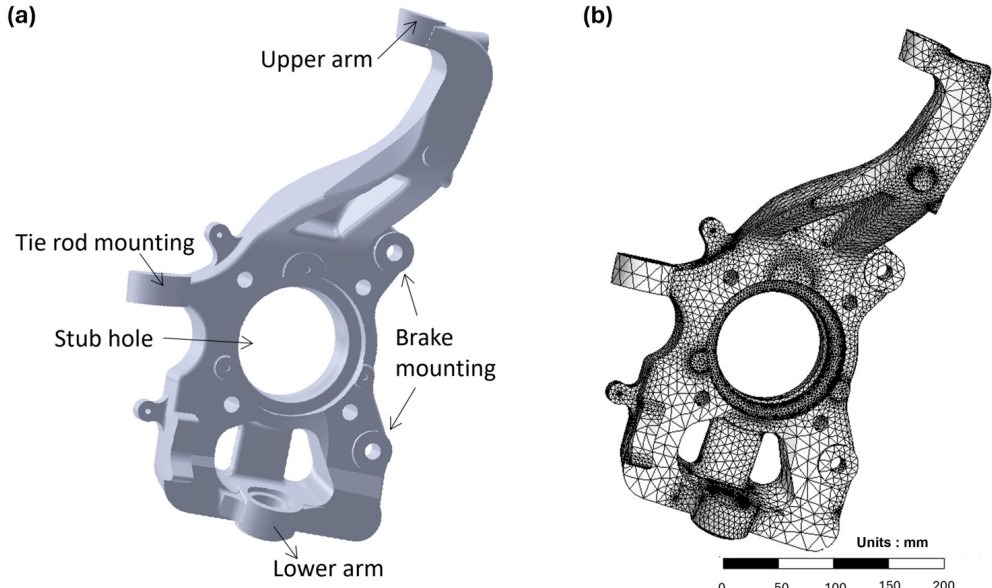

**Figure 1.** (**a**) Steering knuckle characteristics and (**b**) discretization of numerical model.

In Figure 1b, the discretization of the front suspension knuckle is observed. Tetrahedral elements were selected using a nonlinear approach. The mesh consists of 358,973 nodes and 234,227 elements, and has a skewness value of 0.30. The finite element model was validated through the results of a mesh convergence study. Consecutive elements of 0.1 mm, 0.5 mm, 1 mm and 5 mm in size were implemented in order to analyze the numerical model and to determine the adequate mesh size (see Figure 2).

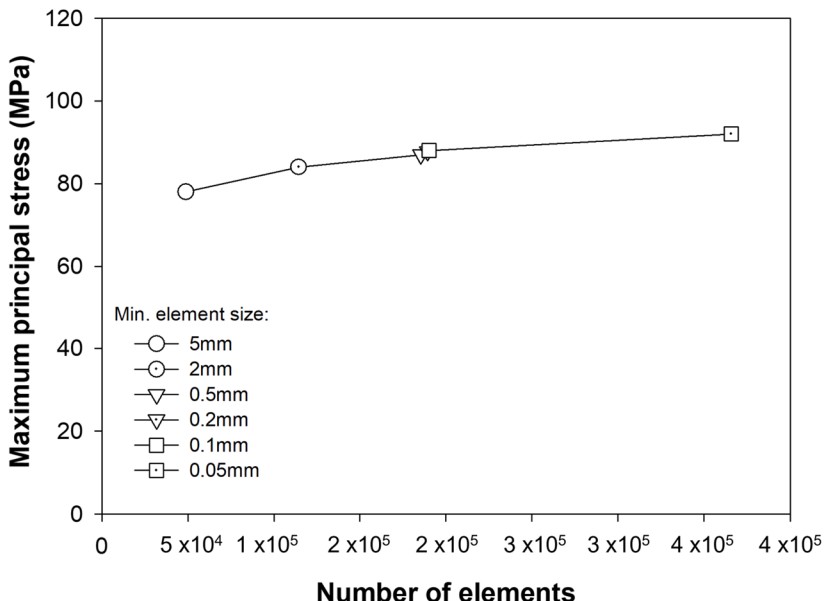

**Figure 2.** Mesh convergence study of numerical model.

Several methods were considered to minimize finite element modeling errors. Mesh refinement enhanced accuracy by reducing the element size in critical areas. Convergence testing ensured stability with increasing refinement. Element type selection considered geometry and behavior for improved accuracy. Accurate material modeling reflected real-world responses and proper boundary conditions prevented unrealistic stress concentrations. These measures collectively enhanced the accuracy and reliability of the finite element simulations.

Table 1 summarizes the mechanical properties of selected materials as alternatives for the steering knuckle component. For Al-6061-T6 alloy, a mechanical test procedure was developed from a commercially available front suspension knuckle. Specimens were manufactured according to the ASTM E8 standard. The specimens were tested at a speed of 0.5 mm/min using an MTS universal testing machine with a capacity of 250 kN. A calibrated extensometer was placed in each test at 25 mm.

**Table 1.** Mechanical properties of materials used for knuckle components.

| Alloy | Yield Strength (MPa) | Elongation (%) | Ultimate Tensile Strength (MPa) | Modulus of Elasticity (GPa) |
|---|---|---|---|---|
| 6092/SiC/17.5p-T6 [25] | 448 | 6 | 510 | 105 |
| 6092/SiC/25p-T6 [26] | 517 | 4 | 538 | 123 |
| Al 6061-T6 [our work] | 276 | 17 | 310 | 68.9 |

The selected materials include an Al MMC 6092/SiC/25p-T6 alloy, which consists of Al 6092 and 25 vol% silicon carbide particles, an Al MMC 6092/SiC/17.5p-T6 with 17.5 vol% silicon carbide particles and a standard Al 6061-T6 alloy. A Poisson ratio of 0.29 was considered for the reinforced Al alloys [26].

The forces acting on the steering knuckle are calculated by empirical relations based on dynamic conditions which are applied by vehicle manufacturers. Table 2 shows the

different dynamic loading conditions in terms of mass (m) and gravity force (g). The braking force, lateral force and steering force were calculated and applied in different locations of the knuckle.

**Table 2.** Dynamic loading conditions of steering knuckle [18].

| Force | Term |
|---|---|
| Braking | 1.5 mg |
| Lateral | 1.5 mg |
| Steering | 50 N |
| Knuckle hub in x-axis | 3 mg |
| Knuckle hub in y-axis | 3 mg |
| Knuckle hub in z-axis | 1 mg |

In order to illustrate the implemented methodology, the calculation of dynamic loads for a Wagon-R Maruti Suzuki vehicle with a mass of 1742 kg (435 kg per wheel) is presented as a reference. Table 3 presents the detailed results of the load calculation for a vehicle with a mass of 1742 kg (class 1). The total mass of the vehicle is divided by four in order to evaluate the load effect on each knuckle.

**Table 3.** Calculus of loads applied on the steering knuckle in class 1 vehicle.

| Loading Condition | Expression |
|---|---|
| Force acting on supports (A, B) | 1.5 mg = 1.5 × 435 kg × 9.81 m/s$^2$ = 6401 N |
| Force on the hub, x-axis | 3 mg = 3 × 435 × 9.81 = 12,802 N |
| Force on the hub, y-axis | 3 mg = 3 × 435 × 9.81 = 12,802 N |
| Force on the hub, z-axis | 1 mg = 1 × 435 × 9.81 = 4267 N |
| Resultant force on the hub (E) | $F = \sqrt{X^2 + Y^2 + Z^2}$ F = 18,600 N |
| Lateral force (F) | =50 N |
| Braking force (C, D) | 1.5 mg = 1.5 × 435 kg × 9.81 m/s$^2$ = 6401 N |

Figure 3 illustrates the location of forces and movement restrictions in different sections of the steering knuckle. The loads are applied to the internal surface of the upper and lower arms, brake mounting, tie rod and stub hole. The stub hole houses the bearing of the wheel and transfers different loads during acceleration, braking and bumping.

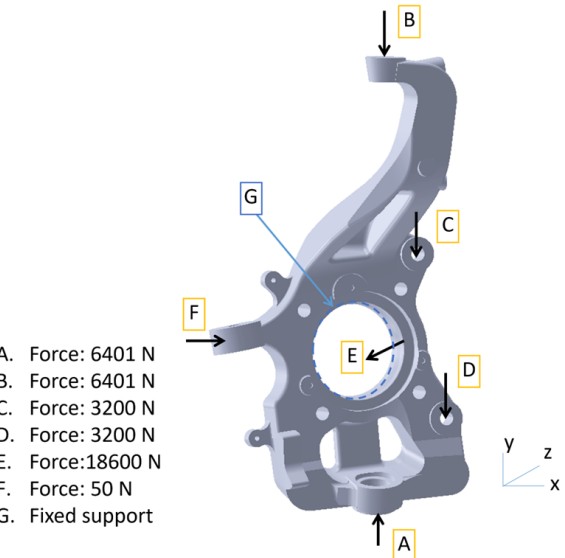

A. Force: 6401 N
B. Force: 6401 N
C. Force: 3200 N
D. Force: 3200 N
E. Force:18600 N
F. Force: 50 N
G. Fixed support

**Figure 3.** Loading conditions of steering knuckle for class 1 vehicle (Wagon-R, Maruti Suzuki).

Four loading scenarios (classes 1–4) are calculated based on the gross weight of the vehicle. In the case of "last-mile" electric vehicles (medium heavy-duty), <12 t, the Gross Vehicle Weight Rating (GVWR) increases up to 30% with respect to a diesel vehicle due to the packaged battery set and components of electrification [1]. The application of forces is distinguished by classes 1 to 4 (see Table 4). The electrification overweighting is specified in a separate column and the forces are calculated based on the total weight (column 1). It is observed that the weight for a class 1 EV is 1742 kg, while the weight for a class 4 EV is 8255 kg.

**Table 4.** Loading conditions of steering knuckles for different EV gross weights.

| Class | GVWR (kg) | Electrification Overweighting (kg) | Force Acting on the Strut (N) | Forces Acting on Knuckle Hub (N) | | |
|---|---|---|---|---|---|---|
| | | | | X, Y | Z | Resultant |
| 1 | 1742 | 402 | 6401 | 12,802 | 4267 | 18,600 |
| 2 | 3538 | 816 | 13,015 | 26,016 | 8672 | 37,800 |
| 3 | 5896 | 1361 | 21,689 | 43,379 | 14,459 | 63,028 |
| 4 | 8255 | 1904 | 30,368 | 60,736 | 20,245 | 88,247 |

## 3. Results and Discussion

The results of the structural analysis for vehicles from classes 1 to 4 are presented in this section. The linear elastic properties described in Figure 1 were applied for the steering knuckle evaluation. Figure 4 shows a comparison of the effective stress and deformation of different conventional knuckle alloys for class 1 vehicles. The numerical results compare two cast iron alloys (commercial products), which are reported in the previous literature, in addition to Al 6061-T6 alloys applied to front steering knuckles. For the validation of the numerical results, a previous work by Kim [11] was verified. The maximum stress reported for the reverse kerb strike load case was 114 MPa. The research aimed to enhance automobile fuel efficiency by reducing weight, focusing on knuckle design. Using 6061 aluminum alloy instead of FCD600 cast iron, stiffness was maintained while reducing weight.

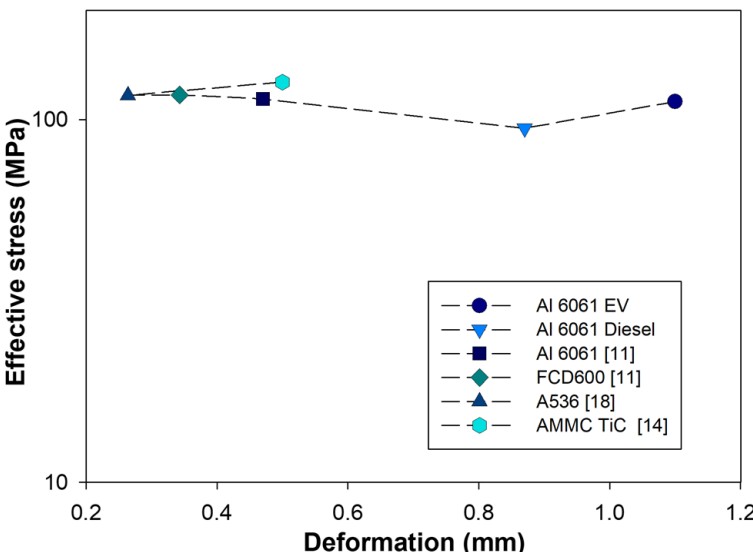

**Figure 4.** Comparison of effective stress (MPa) and deformation (mm) for different steering knuckle alloys in class 1 vehicles.

A maximum effective stress value of about 127 MPa was observed for steering knuckles made of Al-10 wt.% TiC (AMMC TiC) under vertical loading [14]. The minimum stress value was found in the knuckles of Al 6061-T6 alloy for diesel vehicles, which also had

the maximum recorded deformation value. This behavior is associated with the low modulus of elasticity of Al 6061-T6 compared to steels and AMMC TiC. However, the maximum elongation for Al 6061-T6 alloys was about 17%, as reported in Table 1. Under these loading conditions (class 1), the Al 6061-T6 alloys show potential as an alternative material for knuckle components. The numerical results for the Al alloys agree with the previous reported values by Vijayarangan et al. [14], where an Al 6061-T6 alloy was evaluated under standard load cases. The work focused on the automotive industry in general, emphasizing the need for advanced materials and weight reduction in steering knuckle design. However, our work targets heavy-duty last-mile cargo vehicles, addressing challenges related to electrification and battery weight compensation.

Figure 5 presents the numerical results for AMMC 6092/SiC/17.5p-T6 alloy for EVs of class 1 and class 2.

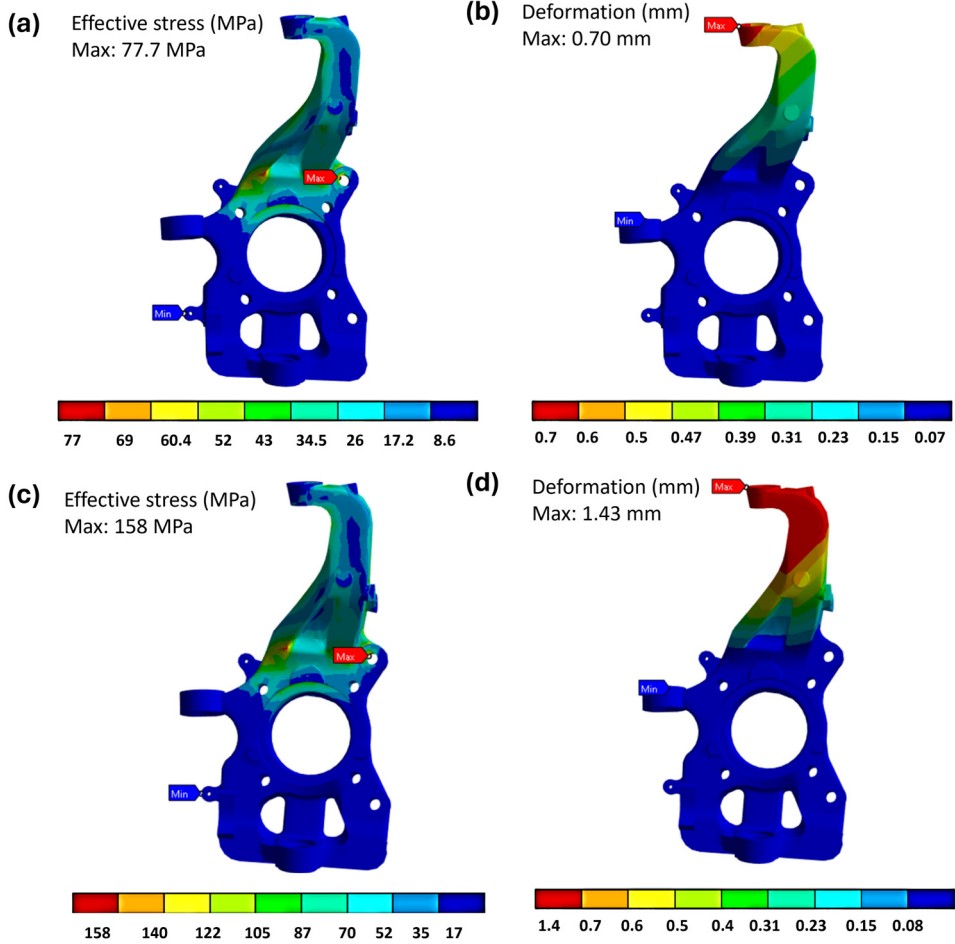

**Figure 5.** Numerical results of steering knuckle 6092/SiC/17.5p-T6 alloy, (**a**) class 1 von Mises stress (MPa), (**b**) class 1 total deformation (mm), (**c**) class 2 von Mises stress (MPa) and (**d**) class 2 total deformation (mm).

The maximum weight for the class 2 EV is considered as 3538 kg, according to Table 3. The numerical results show that a higher stress concentration is found on the upper arm section. This region is critical for the steering knuckle due to its geometrical complexity and variable shape. A maximum value of 158 MPa was determined for the class 2 EV. The maximum deformation occurs in the suspension mounting section with a maximum value of 1.43 mm, while the minimum deformation is found on the stub hole center. The factor of safety for the class 2 EV is 2.66. Figure 6 presents the numerical results of the steering knuckles of class 3 and class 4 EVs (maximum gross weight 8255 kg) with a 6092/SiC/17.5p-T6 alloy.

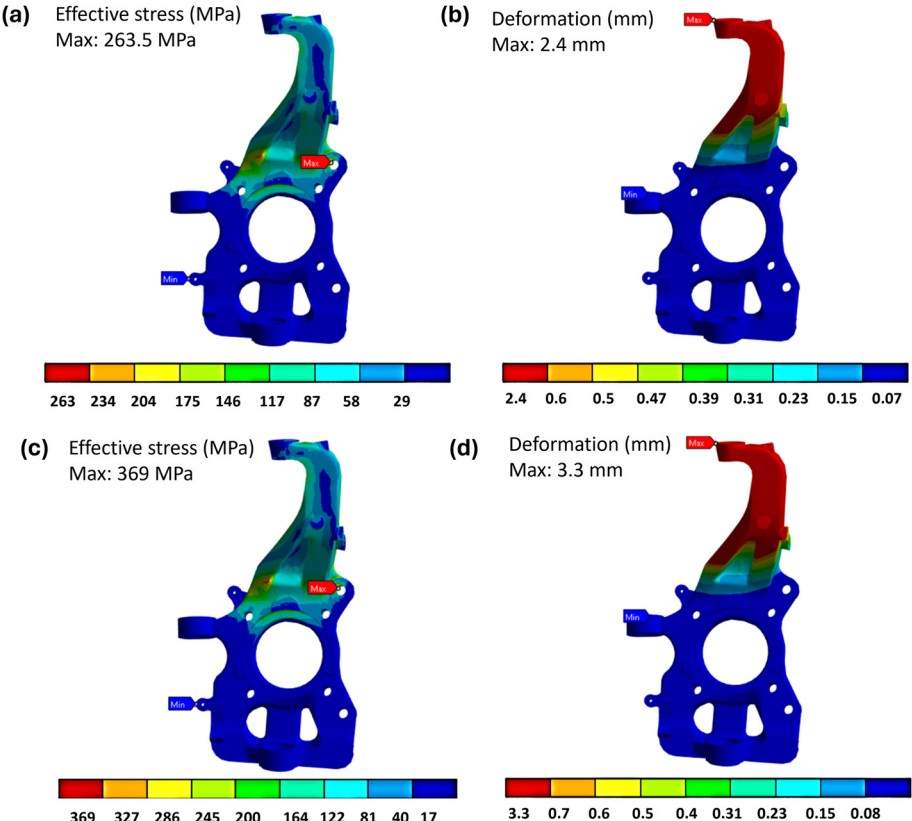

**Figure 6.** Numerical results of steering knuckle 6092/SiC/17.5p-T6 alloy, (**a**) class 3 von Mises stress (MPa), (**b**) class 3 total deformation (mm), (**c**) class 4 von Mises stress (MPa) and (**d**) class 4 total deformation (mm).

The maximum effective stress value was 369 MPa for the class 4 EV, with a factor of safety of 1.2. The maximum stress was located at the root of the suspension mounting section. Moreover, the maximum deformation occurs in the suspension mounting section with a maximum value of 3.3 mm. The maximum elongation of Al reinforced composite alloy is about 6% and the strain at failure is 6.5% [8]. Figure 7 shows a numerical comparison of the change in stress based on the change in loading conditions (class 1–4) for a knuckle made of 6092/SiC/17.5p.

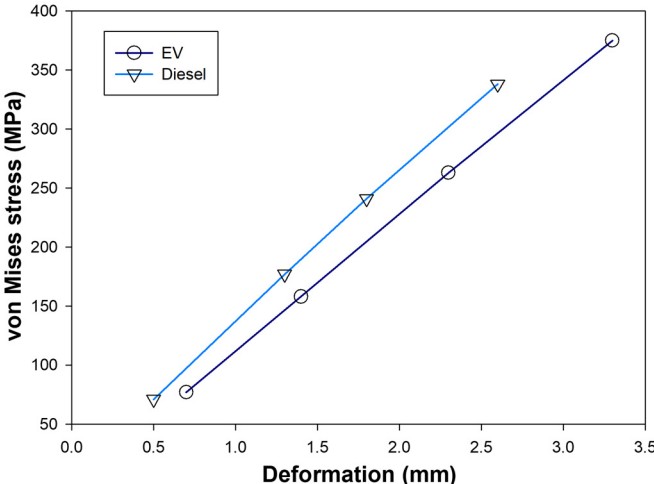

**Figure 7.** Von Mises stress vs. deformation for steering knuckle made of 6092/SiC/17.5p-T6 alloy with different gross weights.

This comparison considered the maximum deformation and maximum effective stress for diesel and electric vehicles with different vehicle gross weights. The maximum stress and deformation values are present in higher vehicle weights. The numerical results indicate that AMMC alloys present a higher effective stress and lower deflection in contrast to the Al 6061-T6 alloy. The ductility of the reinforced composites is reduced due to the formation of precipitates in the Al matrix. However, different heat treatments can be applied to modify the microstructure and mechanical properties of the reinforced composite alloy [8]. Heat treatment affects the interface between the Al matrix and SiC particles, altering the microstructure and fracture behavior. The T6 condition decreases ductility due to intermetallic compound formation. In addition, Reza [24] observed that under static conditions the reliability factor of steering knuckles was higher for reinforced Al alloys in contrast to non-reinforced Al alloys.

The numerical results for the steering knuckles of the class 1 and class 2 EVs reinforced with 25% SiC particles presented a higher stress concentration at the root of the upper arm section. A maximum stress value of 158 MPa was recorded for the class 2 EV, while the maximum deformation occurs in the upper arm with a value of 1.3 mm. Figure 8 presents the numerical results of the class 3 and class 4 vehicle gross weights for the 6092/SiC/25p-T6 alloy.

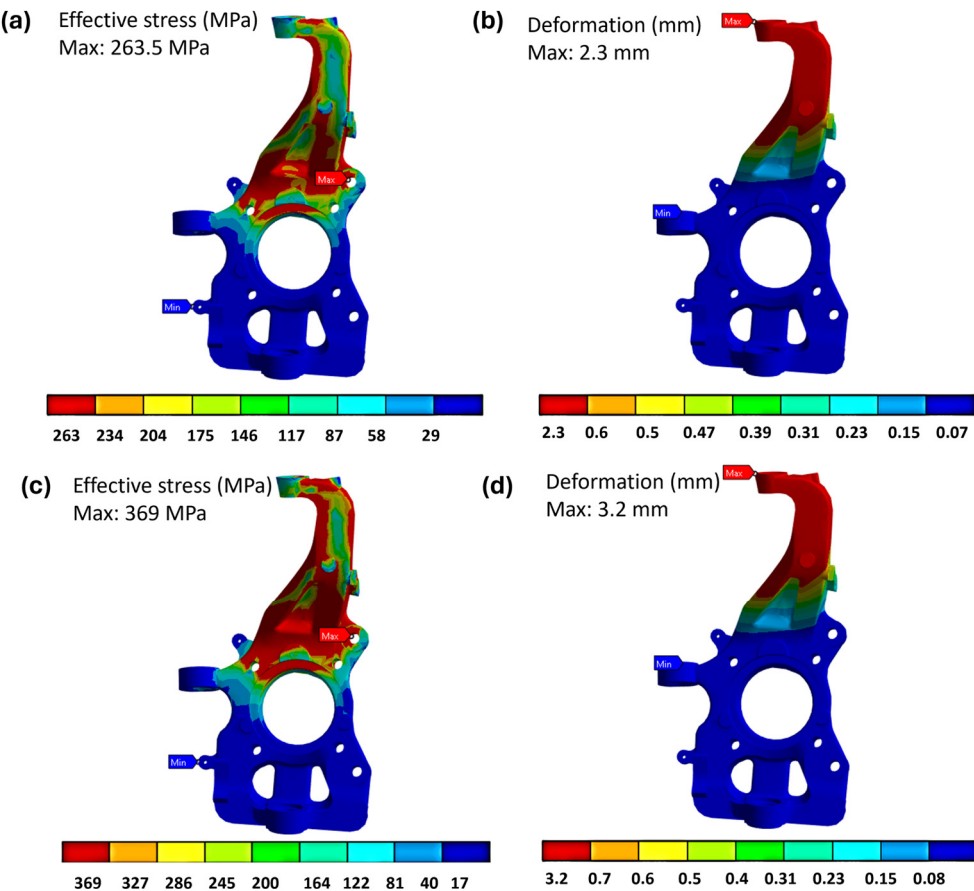

**Figure 8.** Numerical results of steering knuckle 6092/SiC/25p-T6 alloy, (**a**) class 3 von Mises stress (MPa), (**b**) class 3 total deformation (mm), (**c**) class 4 von Mises stress (MPa) and (**d**) class 4 total deformation (mm).

A higher stress concentration was observed for the class 3 and 4 EVs. The maximum stress value of 369 MPa is located at the breaking section of the knuckle. A maximum deformation of 3.2 mm was recorded for the class 4 EV. The maximum elongation for AMMC reinforced with 25% SiC particles is about 4% and the strain at failure is 4.6% [27].

Commercial knuckles are fabricated using cast iron such as FCD600 material [11]. Based on the dimensions of the studied front knuckle, a cast iron knuckle weighs 17,196 g, but this was reduced by about 65% to 6125 g when reinforced Al alloys were utilized. The numerical results for effective stress and deformation for different vehicle gross weights (classes 1 to 4) are shown in Figure 9.

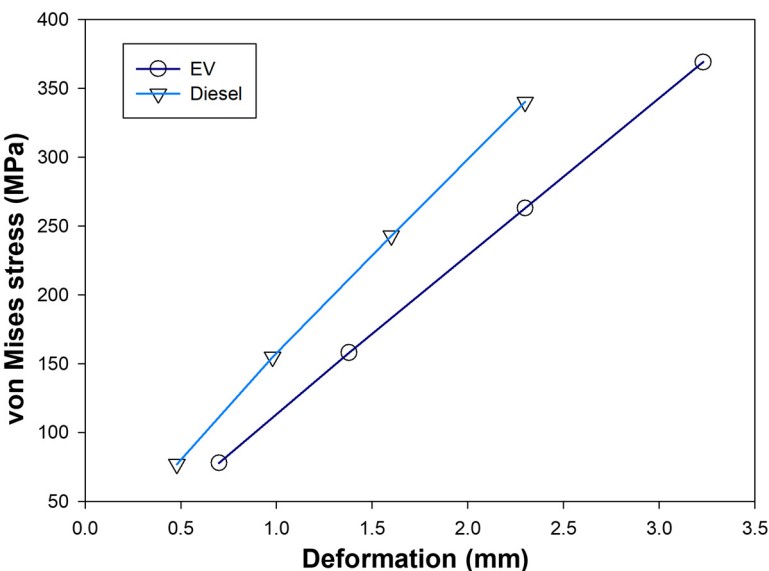

**Figure 9.** Von Mises stress and strain results for different vehicles using a 6092/SiC/25p-T6 alloy.

A comparison of EVs and diesel vehicles is included for the studied loading conditions. The stress results are similar to those of the steering knuckles reinforced with 17.5% SiC particles. However, the deformation is slightly reduced for the reinforced composite with 25% SiC particles. Although the numerical study considers static loading conditions, previous studies reported that the fatigue properties of AMMC alloys increase under axial loading [24]. The target life based on industrial requirements is specified as $1 \times 10^6$ cycles [14]. The fatigue strength of reinforced composites with 17.5% SiC particles is higher than that of Al 6061-T6, according to Harringan [27]. Powder metallurgy consistently yields high-property composites, with recent efforts aimed at cost reduction. Figure 10 shows a comparison of Al 6061-T6 knuckle alloys versus MMC alloys for the different classes of vehicles. It is observed from the numerical results that MMC alloys show potential as an alternative material for class 3 and class 4 vehicles based on the stress and deformation values.

The maximum elongation for AMMC alloys lies between 4 and 6%, where the principal mechanism influencing ductility is associated with void nucleation and coalescence of the SiC particles. Ductile fractures are observed at low strain rates, while interfacial cracks occur under high strain rates due to the elevated stress levels [8]. The maximum von Mises stress for class 4 AMMC alloys is above 350 MPa. This condition is limited due to the fracture toughness of the reinforced alloys. Li et al. [28] investigated the failure mechanism of 6092 Al reinforced with 17.5% and 25% SiC particles. It was found that the threshold stress for particle cracking becomes a dominant failure mechanism at about 350 MPa, affecting the ability to carry load. In addition, the Al 6061-T6 alloys presented a stress value above 300 MPa, which limits their application for higher gross weight vehicles (class 3 and class 4). While the knuckle geometry is expected to change for different cargo vehicles, this work aims to evaluate the feasibility of use AMMC alloys in critical components such as steering knuckles and to propose solutions to improve conventional vehicle suspension and overcome associated challenges. This study is expected to be applied as a lightweight design guide that compares the change in stresses according to the change in the load conditions of different cargo vehicles.

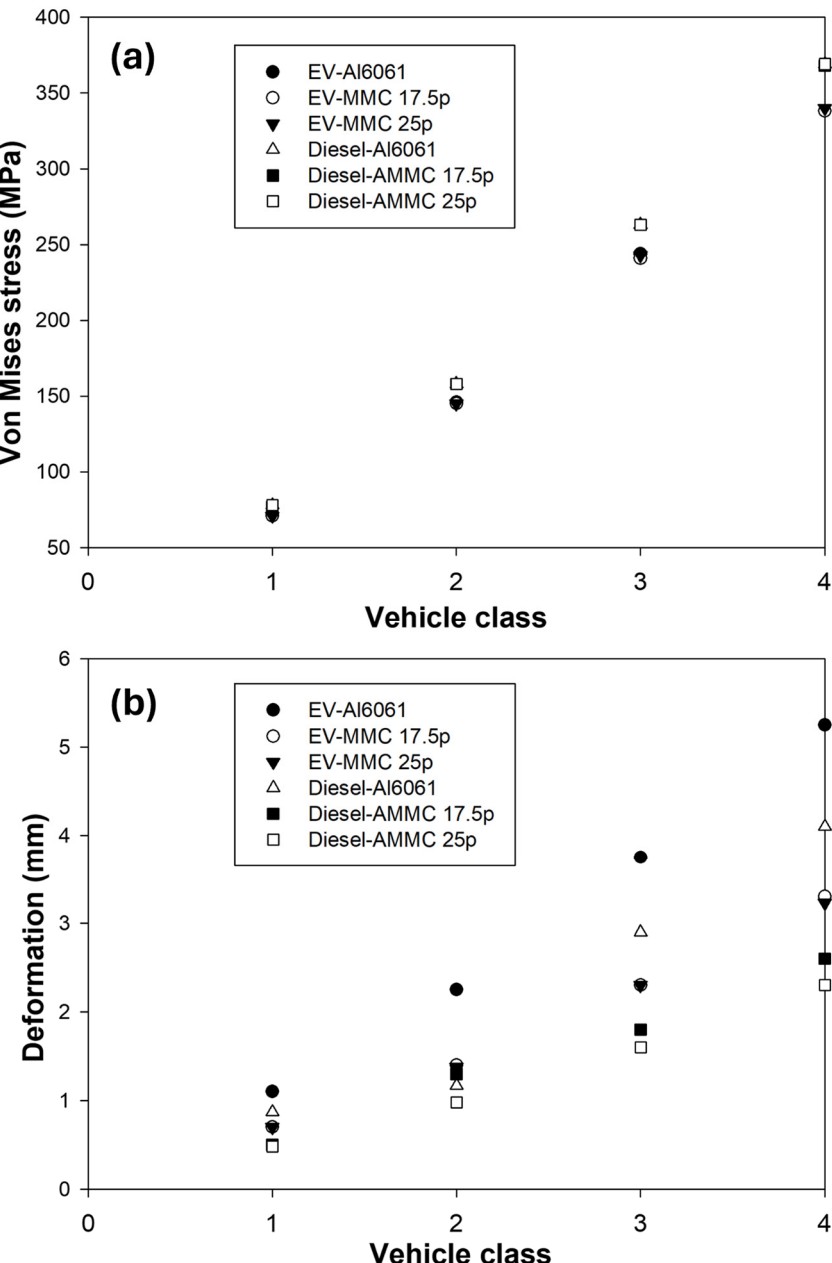

**Figure 10.** Numerical results of aluminum alloys for steering knuckles, (**a**) von Mises stress (MPa) and (**b**) maximum deformation (mm).

## 4. Conclusions

A numerical study was developed for the mechanical evaluation of steering knuckle components reinforced with SiC particles. Two different percentages of silicon carbide particles were investigated in steering knuckles of medium-duty electrical vehicle configurations (class 1–4). The relevant achievements of this work are as follows:

- The critical section of the steering knuckle is the root of the upper arm.
- The comparison of deformation for Al 6061-T6 versus MMC alloys (SiC-17.5 and SiC-25) shows a decrease in deformation close to 50%; this reduction was 2.3 mm for AMMC with 25% SiC particles.
- The reinforced composite alloys present a lower effective stress than the yield stress for the different classes of vehicles studied. However, the high stress (369 MPa) and moderate ductility of 6092 Al reinforced composites limits its application in class 4 vehicles.

- For lower gross weight vehicles, AMMC alloys are potential alternatives for the steering knuckles of electrified cargo vehicles.
- The weight reduction is about 65% when compared with commercial cast iron. This study has provided a significant contribution to the lightweight design of steering knuckles for last-mile cargo vehicles through rigidity reinforcement implementing AMMC alloys.

**Author Contributions:** Conceptualization, J.O.-H. and L.H.; methodology, C.S.; software, A.R.; formal analysis, P.Z.-R.; investigation, L.R.-O.; writing—original draft preparation, L.R.-O.; writing—review and editing, J.O.-H. All authors have read and agreed to the published version of the manuscript.

**Funding:** The authors of this work are grateful for the support of the ProActi project of University of Nuevo Leon.

**Data Availability Statement:** The original contributions presented in the study are included in the article, further inquiries can be directed to the corresponding author.

**Conflicts of Interest:** Jesus Orona-Hinojos, Lizbeth Huerta and Alfredo Rios are employees of DataSc Consultancy Group. The paper reflects the views of the scientists, and not the company.

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
