# Peer review of "Numerical Study of Reinforced Aluminum Composites for Steering Knuckles in Last-Mile Electric Vehicles"

_wevj, doi:10.3390/wevj15030109_

Round 1

Reviewer 1 Report

Comments and Suggestions for Authors

The authors introduced numerical study of reinforced aluminum composites for steering knuckles in last-mile electric vehicles The study systematically presents a numerical methodology to 14 evaluate aluminum metal matrix composites (AMMC), series 60XX with reinforcement ceramics 15 particles, for steering knuckle component in heavy-duty last mile cargo vehicle.

1.      Abstract should be given as numericaÅŸ values.

2.      Research gap studies are still need to be elaborate more.

3.      Novely is still needed to be emphasized well.

4.      The work literature is very interesting. Its good to see it.

5.       Given that the manuscript is heavily based on simulational work and measurements, The methods/measures that have been taken to minimize modelling errors.

6.      Further, results and analysis of finite element were still not compared other studies well

7.      Please proofread the paper again.

Comments on the Quality of English Language

Minor editing of English language required

Author Response

ID (wevj-2788375)

Feb-27-2024

Dear Editor

We are grateful to the editorial board and reviewer for the consideration and constructive comments on our manuscript. We have addressed the suggestions of the reviewer and wish to submit a revised version of the manuscript for further consideration in the journal.

Changes in the updated version of the manuscript are highlighted. Below, we also provide a point-by-point response explaining how we have addressed each of the reviewer’s comments.

We look forward to the outcome of your assessment.

Yours sincerely,

On behalf of the co-authors

Dr. Luis Reyes

Comments and Suggestions for Authors

The authors introduced numerical study of reinforced aluminum composites for steering knuckles in last-mile electric vehicles The study systematically presents a numerical methodology to 14 evaluate aluminum metal matrix composites (AMMC), series 60XX with reinforcement ceramics 15 particles, for steering knuckle component in heavy-duty last mile cargo vehicle.

  1. Abstract should be given as numericaÅŸ values.

OUR ANSWER: Thank you for this comment. The required changes have been implemented.

The steering knuckle is a critical component of suspension and steering drive systems of electric vehicles. Electrification of last-mile vehicles presents a challenge in terms of cost, driving range and compensation of battery weight. This work presents a numerical methodology to evaluate 60XX series aluminum metal matrix composites (AMMC) with reinforcement ceramics particles for steering knuckle components in heavy-duty last mile cargo vehicle. The use of AMMC provides light weight knuckles with sufficient strength, stiffness and safety conditions for electrical vehicle cargo configurations. The numerical study includes three aluminum alloys, two AMMC alloys and an Al 6061-T6 alloy as reference materials. Medium Duty Heavy Vehicle Class <12t, such as electrical vehicle cargo configurations, are considered for the numerical study (class 1-4). The maximum von Mises stress for class 4 AMMC alloys exceeds 350 MPa, limited by fracture toughness. The weight reduction is about 65% when compared with commercial cast iron. Moreover, Al 6061-T6 alloys exhibit stress values surpassing 300 MPa, constraining their suitability for heavier vehicles. The study proposes assessing the feasibility of implementing AMMC alloys in critical components like steering knuckles and suggests solutions to enhance conventional vehicle suspension systems and overcome associated challenges. It aims to serve as a lightweight design guide, offering insights into stress variations with differing load conditions across various cargo vehicles.

  1. Research gap studies are still need to be elaborate more.

OUR ANSWER: Thank you for this comment. The required changes have been implemented.

The research highlights several significant research gaps in the field of AMMC alloys for steering knuckles, particularly in the context of electrified cargo vehicles:

  1. Lack of Research on AMMC Alloys for Steering Knuckles: Despite existing research on AMMC alloys, there is a notable lack of studies specifically focused on their application in steering knuckles. This gap indicates a need for investigation into the suitability and performance of AMMC alloys specifically tailored for steering knuckles.
  2. Absence of Studies on Electrified Cargo Vehicles: The research gap extends to the absence of studies evaluating AMMC alloys for steering knuckles of electrified cargo vehicles. Given the unique requirements and operating conditions of electrified vehicles, there is a need for research to assess the viability and effectiveness of AMMC alloys in this context.
  3. Limited Understanding of Mechanical Properties and Forces: The significance of understanding the mechanical properties and forces that steering knuckles can withstand is underscored. This gap suggests a need for comprehensive investigations into the strength, durability, and performance characteristics of steering knuckles, particularly when reinforced with lightweight AMMC alloys.
  4. Lack of Numerical Studies on Dynamic Loading Conditions: Another research gap pertains to the scarcity of numerical studies considering various dynamic loading conditions, such as braking force, lateral force, and steering force, applied to steering knuckles. Understanding the response of steering knuckles under different loading scenarios is essential for optimizing their design and performance.
  5. Focus on Specific Vehicle Classes: The research focuses on medium-duty heavy vehicle classes (<12t), particularly electrified cargo vehicle configurations (class 1-4). This indicates a specific gap in research pertaining to the application of AMMC alloys in steering knuckles for these vehicle classes, highlighting the need for targeted investigations tailored to their requirements.

Addressing these research gaps through numerical studies and static stress analyses can facilitate the development of lightweight designs, propose solutions to enhance conventional vehicle suspension systems, and overcome associated challenges in the context of electrified cargo vehicles.

  1. Novely is still needed to be emphasized well.

OUR ANSWER: Thank you for this comment. The required changes have been implemented.

The novelty of this work lies in its comprehensive approach to evaluating the strength of front steering knuckles reinforced by aluminum metal matrix composites through numerical analysis. Unlike previous research, this study considers various dynamic loading conditions, including braking force, lateral force, and steering force, applied at different locations on the knuckle. Particularly groundbreaking is the focus on Medium Duty Heavy Vehicle Class configurations (<12t), specifically electrified vehicle cargo setups (class 1-4), which have received limited attention in existing literature. Additionally, the incorporation of static stress analysis enables the investigation of lightweight designs and the proposal of solutions to enhance conventional vehicle suspension systems, addressing associated challenges. This holistic examination of AMMC-reinforced steering knuckles in the context of electrified cargo vehicles represents a novel contribution to the field, offering insights into optimizing performance and durability while advancing lightweight design principles.

  1. The work literature is very interesting. Its good to see it.

OUR ANSWER: Thank you for this comment. The required changes have been implemented.

  1. Given that the manuscript is heavily based on simulational work and measurements, The methods/measures that have been taken to minimize modelling errors.

OUR ANSWER: Thank you for this comment. The required changes have been implemented.

Several methods were considered to minimize finite element modeling errors. Mesh refinement enhanced accuracy by reducing element size in critical areas. Convergence testing ensured stability with increasing refinement. Element type selection considered geometry and behavior for improved accuracy. Accurate material modeling reflected real-world responses. Proper boundary conditions prevented unrealistic stress concentrations. These measures collectively enhanced the accuracy and reliability of finite element simulations.

  1. Further, results and analysis of finite element were still not compared other studies well

OUR ANSWER: Thank you for this comment. The required changes have been implemented.

…The numerical results for Al alloys agree with the previous reported values by Vijayarangan et al. [14], where an Al 6061-T6 alloy was evaluated under standard load cases. The work focused on the automotive industry in general, emphasizing the need for advanced materials and weight reduction in steering knuckle design. However, our work targets heavy-duty last-mile cargo vehicles, addressing challenges related to electrification and battery weight compensation.

…The maximum stress reported for the reverse kerb strike load case was 114 MPa. The research was aimed to enhance automobile fuel efficiency by reducing weight, focusing on knuckle design. Using 6061 aluminum alloy instead of FCD600 cast iron, stiffness was maintained while reducing weight.

…Heat treatment affects the interface between Al-matrix and SiC particles, altering microstructure and fracture behavior. T6 condition decreases ductility due to intermetallic compound formation.

The powder metallurgy consistently yields high-property composites, with recent efforts aimed at cost reduction. Figure 10 shows a comparative of Al 6061-T6 knuckle alloys versus MMC alloys for different classes.

.. . Ductile fractures are observed at low strain rates, while interfacial cracks occur under high strain rates due to elevated stress levels [8].

  1. Please proofread the paper again.

Comments on the Quality of English Language

Minor editing of English language required

OUR ANSWER: Thank you for this comment. The required changes have been implemented.

Reviewer 2 Report

Comments and Suggestions for Authors

I have received responses to most of my comments in the previous review round of this article. The authors introduced changes very superficially. However, progress in improving the article is visible. Unfortunately, there are still inaccuracies in the article, which in terms of material properties are greater in this version of the manuscript than in the previous one.

My comment:
"The values given in Table 1 do not coincide with the values presented in [20]. The authors used material data in their analyses that are neither experimentally nor literature confirmed."
was not resolved.

a) Table 1: it has been identified that data for 6092/SiC/25p-T6 was taken from [26]. The yield stress 448 in [26] corresponds to the MMCs consisted particulate vol.% is equal to 17.5. The correct yield stress is equal to 517 MPa. So, the numerical results presented in this article were performed on the wrong data.

b) Table 1: it has been identified that data for Al 6061-T6 were taken from [11]. According to [11] the yield stress for AA6061-T6 aluminium alloy is 270 MPa. The authors used the value of 276 MPa in the numerical models. Moreover, elongation is different in [11] and manuscript reviewed. Elongation is not necessary in the numerical model, however, the correctness of data in a scientific article is required.

c) Table 1: it has been identified that data for Al 6061-T6 were taken from [11]. According to [11] the modulus of elasticity of AA6061-T6 aluminium alloy is 71 GPa. The authors used the value of 68.9 MPa in the simulations. To sum up, the numerical modeling uses data that is not confirmed by literature sources or own research. Therefore, the numerical results cannot be considered reliable.

There are still many mixed styles of designation of aluminium alloys.

Author Response

ID (wevj-2788375)

Feb-27-2024

Dear Editor

We are grateful to the editorial board and reviewer for the consideration and constructive comments on our manuscript. We have addressed the suggestions of the reviewer and wish to submit a revised version of the manuscript for further consideration in the journal.

Changes in the updated version of the manuscript are highlighted. Below, we also provide a point-by-point response explaining how we have addressed each of the reviewer’s comments.

We look forward to the outcome of your assessment.

Yours sincerely,

On behalf of the co-authors

Dr. Luis Reyes

Comments and Suggestions for Authors

I have received responses to most of my comments in the previous review round of this article. The authors introduced changes very superficially. However, progress in improving the article is visible. Unfortunately, there are still inaccuracies in the article, which in terms of material properties are greater in this version of the manuscript than in the previous one.

My comment:
"The values given in Table 1 do not coincide with the values presented in [20]. The authors used material data in their analyses that are neither experimentally nor literature confirmed."
was not resolved.

a) Table 1: it has been identified that data for 6092/SiC/25p-T6 was taken from [26]. The yield stress 448 in [26] corresponds to the MMCs consisted particulate vol.% is equal to 17.5. The correct yield stress is equal to 517 MPa. So, the numerical results presented in this article were performed on the wrong data.

OUR ANSWER: Thank you for this comment.

A correction has been made to the text. In the context of an elastic material, the yield stress does not directly affect the finite element analysis (FEA) because the material remains within the elastic regime where deformation is reversible and no permanent deformation occurs. However, the yield stress becomes relevant when the material exceeds its elastic limit and enters the plastic regime, at which point the material undergoes permanent deformation. Therefore, in FEA of elastic materials, the focus is on parameters such as Young's modulus and Poisson's ratio, which describe the material's elastic behavior.

  1. b) Table 1: it has been identified that data for Al 6061-T6 were taken from [11]. According to [11] the yield stress for AA6061-T6 aluminium alloy is 270 MPa. The authors used the value of 276 MPa in the numerical models. Moreover, elongation is different in [11] and manuscript reviewed. Elongation is not necessary in the numerical model, however, the correctness of data in a scientific article is required.

OUR ANSWER: Thank you for this comment. The required changes have been implemented.

For Al- 6061-T6 alloy, a mechanical test procedure was developed from a commercially available front suspension knuckle.

This data fairly agrees with previous reported literature.

Hellier, A. K., Chaphalkar, P. P. & Prusty, B. G. (2017). Fracture toughness measurement for aluminium 6061-T6 using notched round bars. 9th Australasian Congress on Applied Mechanics (ACAM9) (pp. 332-339). Sydney: Engineers Australia.

  1. c) Table 1: it has been identified that data for Al 6061-T6 were taken from [11]. According to [11] the modulus of elasticity of AA6061-T6 aluminium alloy is 71 GPa. The authors used the value of 68.9 MPa in the simulations. To sum up, the numerical modeling uses data that is not confirmed by literature sources or own research. Therefore, the numerical results cannot be considered reliable.

OUR ANSWER: Thank you for this comment. The required changes have been implemented.

For Al- 6061-T6 alloy, a mechanical test procedure was developed from a commercially available front suspension knuckle.

There are still many mixed styles of designation of aluminium alloys.

OUR ANSWER: Thank you for this comment. The required changes have been implemented.

Reviewer 3 Report

Comments and Suggestions for Authors

The text submitted for re-evaluation may already be published. the final decision is made by the chief editor.

Author Response

ID (wevj-2788375)

Feb-27-2024

Dear Editor

We are grateful to the editorial board and reviewer for the consideration and constructive comments on our manuscript. We have addressed the suggestions of the reviewer and wish to submit a revised version of the manuscript for further consideration in the journal.

Changes in the updated version of the manuscript are highlighted.

We look forward to the outcome of your assessment.

Yours sincerely,

On behalf of the co-authors

Dr. Luis Reyes

Round 2

Reviewer 2 Report

Comments and Suggestions for Authors

The authors introduced changes to the manuscript as suggested by the reviewer. I suggest accepting this manuscript in present form.